# Gender differences in the impact of fatigue on lower limb landing biomechanics and their association with anterior cruciate ligament (ACL) injuries: A systematic review and meta-analysis

Chengxun Liu[1], Wuwen Peng[1], Wenhao Qu[1], Zhiyong Zhang[2], Jian Sun[2,3], Jiaxin He[2*], Bojin Cheng[4*], Duanying Li[2,3*]

1 Graduate School, Guangzhou Sport University, Guangzhou, Guangdong, China, 2 School of Athletic Training, Guangzhou Sport University, Guangzhou, China, 3 Guangdong Provincial Key Laboratory of Human Sports Performance Science, Guangzhou, China, 4 School of Physical Education, Guangzhou Sport University, Guangzhou, China

☉ These authors contributed equally to this work.
* 574401513@qq.com (JH); bobo791017@163.com (BC); liduany@gzsport.edu.cn (DL)

## Abstract

### Background

This meta-analysis examines the impact of neuromuscular fatigue on gender differences in lower limb landing biomechanics and its correlation with ACL injury risk.

### Methods

A comprehensive search was conducted in PubMed, Scopus, Web of Science, Embase, and the Cochrane Library up to March 2024.

### Results

Fourteen studies were included, averaging a quality score of 6.79; nine were high quality. Key findings: males showed a significant increase in knee flexion angle at initial contact (effect size -1.23), but females did not (-0.25). Both genders had significant changes in hip external rotation (males: 1.35, females: 1.20). Ankle peak dorsiflexion angle increased (-1.69) with no gender differences. Peak Knee extension moment increased in males (0.76) and females (0.48) with an overall effect size of 0.64, but no change in peak abduction moment. Peak Hip extension moment was significant in males (0.58) and overall (0.51), with no changes in internal rotation or adduction moments. Peak vertical ground reaction force showed no significant changes for either gender.

### Conclusions

Fatigue alters knee biomechanics in males, raising ACL injury risk, and both genders show increased hip and ankle loads post-fatigue. These results suggest the need for

**Data availability statement:** All relevant data are within the paper and its Supporting information files.

**Funding:** This study was funded by the Guangdong Provincial Philosophy and Social Sciences Regularization Project 2022 (GD22CTY09): Research on the Coordinated Development Path of International Competitiveness in Sports in the Guangdong-Hong Kong-Macao Greater Bay Area (to JS).

**Competing interests:** The authors have declared that no competing interests exist.

gender-specific fatigue management strategies to mitigate ACL injury risk and call for further research into prevention mechanisms.

## Introduction

Anterior Cruciate Ligament (ACL) tears are among the most common non-contact injuries in many sports [1,2], especially those involving rapid direction changes, deceleration, jumping, and landing [3–5]. ACL injuries typically occur at the knee joint due to improper landing techniques and movement patterns, which can place excessive shear forces and stress on the ACL [6]. Changes in these biomechanical characteristics are closely linked to ACL injuries, particularly increases in knee abduction angles, knee abduction moments, and hip internal rotation moments during landing, which are key factors in non-contact ACL injuries [7].

Multiple factors cause biomechanical changes in the lower limbs during landing [8–10], with fatigue being a critical one. Fatigue alters the biomechanical characteristics of the lower limbs during landing, increasing the risk of ACL injuries [11–13]. Fatigue, as a complex physiological mechanism, occurs at both central and peripheral levels, significantly affecting neuromuscular pathways [14]. Under fatigue, reduced neural feedback efficiency, prolonged muscle reaction time, and proprioceptive dysfunction slow down muscle strength generation and contraction speed [15,16]. These physiological changes not only affect kinematic performance [17,18] but also weaken joint stability [19,20], significantly impairing dynamic balance control [21]. Research further supports this, showing that vertical ground reaction forces (VGRF) significantly increase under fatigue, leading to higher knee joint loads and thus a greater risk of ACL injuries [22]. Moreover, fatigue reduces knee flexion angles, increases knee abduction angles, and heightens hip internal rotation angles, significantly raising stress on the knee joint and ACL [23,24]. Additionally, fatigue-induced reduction in knee flexion may relax the medial collateral ligament (MCL) and lateral collateral ligament (LCL), leading to rotational knee instability and further increasing the risk of ACL injury [25]. Epidemiological studies also show that most sports injuries occur in the latter stages of activities or competitions, further underscoring the importance of fatigue in sports injuries [26]. In summary, fatigue increases the risk of ACL injuries by altering lower limb biomechanics through changes in muscle and nervous system functions.

Studies have qualitatively shown gender differences in lower limb landing mechanics [27,28]. For instance, Hewett et al. found that females have smaller knee flexion angles and larger knee valgus angles during landing, leading to a significantly higher ACL injury rate than males [4]. Additionally, insufficient hip flexion and increased internal rotation can indirectly increase knee joint load [29]. Ford et al. found that female athletes have larger hip internal rotation angles during jumping and landing, making them more prone to ACL injuries [30]. However, findings on gender differences in lower limb landing mechanics under fatigue conditions are inconsistent. Fagenbaum et al. [31] employed an isokinetic machine to test six male and eight female basketball players under fatigue. Results showed that fatigued

females had greater knee flexion acceleration during landing than males, who also exhibited changes in muscle coordination, unlike females. This lack of adaptation may increase the risk of ACL injuries in female athletes during jump landings. In contrast, Susan L. Rozzi et al. found that although fatigue decreased joint proprioception (especially in extension) and altered muscle activation patterns, including delayed contraction times and increased specific muscle contraction areas, there were no significant gender differences [32]. These discrepancies indicate the need for more systematic research to explore the impact of fatigue on gender differences in lower limb landing biomechanics.

Therefore, this study aims to systematically review and synthesize existing research through a meta-analysis to explore the specific impact of fatigue on gender differences in lower limb landing biomechanics. This will provide a deeper understanding and scientific basis for preventing ACL injuries, helping to identify high-risk populations and offering references for developing more personalized prevention and rehabilitation programs, thereby effectively reducing the incidence of ACL injuries.

## Methods

This article is prepared according to the PRISMA project guidelines and the recommendations by Moher et al. for systematic reviews and meta-analyses [33]. The protocol has been registered on PROSPERO (ID: CRD42024545104).

### Search strategy

As of March 2024, a comprehensive search was conducted in the electronic databases of PubMed, Scopus, Web of Science, Embase, and Cochrane Library. Boolean operators "AND" and "OR" were used with the following keywords (using Pubmed as an example) (Table 1).

The systematic search was conducted by two independent researchers (C.L., W.Q.). Initially, articles were screened based on titles and abstracts. If the information was unclear or ambiguous, the full texts were retrieved for further assessment. Following this, the researchers compared their findings and resolved any discrepancies through consensus discussions. In cases of unresolved disagreement, a third independent reviewer (W.P.) was consulted to provide a final decision.

To ensure agreement among the investigators, a rigorous process was followed:

1. Predefined Eligibility Criteria: All researchers applied a set of clearly defined inclusion and exclusion criteria, which were established prior to the screening. These criteria covered study design, population characteristics, interventions, and outcomes.

2. Independent Screening: Each researcher independently screened the articles at both the title/abstract and full-text stages based on the predefined criteria.

3. Consensus Process: After the initial screening, any disagreements regarding study eligibility were addressed through consensus discussions. Studies that could not be agreed upon were jointly reassessed by all researchers to reach a final decision.

**Table 1. PubMed literature selection strategy.**

| Query number | Search Terms |
| --- | --- |
| #1 | "fatigue" [Title/Abstract] OR "tiredness" [Title/Abstract] OR "weariness" [Title/Abstract] OR "exhaustion" [Title/Abstract] OR "induced fatigue" [Title/Abstract] OR "fatigue damage" [Title/Abstract] OR "neuromuscular fatigue" [Title/Abstract] |
| #2 | "sexuality" [Title/Abstract] OR "sex" [Title/Abstract] OR "gender" [Title/Abstract] |
| #3 | "landing biomechanics" [Title/Abstract] OR "kinematics" [Title/Abstract] OR "kinetics" [Title/Abstract] OR "landing" [Title/Abstract] OR "Biomechanics" [Title/Abstract] OR "lower extremity" [Title/Abstract] OR "hip" [Title/Abstract] OR "knee" [Title/Abstract] OR "ankle" [Title/Abstract] |
| #4 | #1 AND #2 AND #3 |

4. Third-party Review: When discrepancies persisted or were particularly complex, a third independent reviewer (W.P.), with expertise in the subject area, was consulted to resolve the issue.

5. Quality Assessment Using STROBE Guidelines: The quality of the included studies was evaluated using a modified version of the STROBE guidelines, which assessed key methodological aspects such as study design, data collection, and potential biases.

## Inclusion and exclusion criteria

The literature screening criteria for this meta-analysis were established according to the PICOS [34] format (Participants, Interventions, Comparisons, Outcomes, and Study Design) used in evidence-based medicine. The inclusion criteria are: (1) studies involving both male and female participants; (2) studies utilizing a pre-post self-controlled experimental design; (3) studies implementing at least one protocol designed to induce fatigue; and (4) studies with outcomes that include kinematic, kinetic, or electromyographic analyses relevant to sports biomechanics. The exclusion criteria are: (1) studies not employing a self-controlled experimental design; (2) studies with inaccessible data; (3) studies not utilizing three-dimensional motion analysis systems or force platforms for kinematic and kinetic measurements; (4) conference papers, review articles, and opinion pieces; and (5) studies where the full text is not available.

## Data extraction

All references were managed using EndNote X9 software, and duplicates were removed. Two authors (C.L and W.Q) independently extracted descriptive information from all included studies, including publication details (author, year), participant demographics (age, gender), sample size, participant characteristics, fatigue intervention protocols, implementation methods, and biomechanical outcomes (kinematics, kinetics) (Table 2). Biomechanical data necessary for calculating effect sizes (mean and standard deviation) were extracted, and corresponding authors were contacted for additional data if needed. Any discrepancies in the data extracted by C.L and W.Q were confirmed by a third researcher, W.P.

**Table 2. Checklist of 10 criteria based on an adapted version of the STROBE guidelines for assessing the quality of observational studies used in this analysis.**

| Criteria | Score |
|---|---|
| Materials and methods | |
| 1.   Describe the study setting or participating locations | 1 |
| 2.   Describe the period of recruitment/follow-up/data collection | 1 |
| 3.   Describe the sources/method of selection of participants | 1 |
| 4.   Give the inclusion and exclusion criteria | 1 |
| Data sources/Measurement | |
| 5.   Clearly describe all outcome measures: | 1 |
| 6.   Describe measurement/testing procedures of each outcome measure | 1 |
| 7.   Describe comparability of assessment between groups/time-points | 1 |
| 8.   Explain number of participants with missing data and how this was addressed | 1 |
| 9.   Report number of individuals in each group/time-point (and if applicable any reasons for non-participation in ≥ 1 outcome measure) | 1 |
| 10.  Report means for each fatigue/sex grouping with a measure of variance (e.g., 95% CI or SD) | 1 |
| Total Score | 10 |

## Risk of bias assessment

The quality of the studies was assessed using a modified version of the Strengthening the Reporting of Observational Studies in Epidemiology (STROBE) guidelines for evaluating observational studies [35]. Each included study was scored based on 10 specific criteria derived from items 5, 6, 7, 8, 9, 12, 13, 14, and 15 of the original checklist (Table 3). The overall quality score was the sum of these criteria, with a maximum score of 10 points. Studies scoring 7 points or higher were considered high quality. This scoring threshold was determined by author consensus and based on previously published modifications of the STROBE guidelines [27].

## Data analysis

To ensure objectivity and robustness, Review Manager 5.3 software (The Nordic Cochrane Centre, Copenhagen, Denmark) was utilized for meta-analyses and subgroup analyses of data from at least two studies reporting on the same variable for both males and females. When measurement units were consistent (e.g., angles), the weighted mean difference (WMD) with a 95% confidence interval was used to assess the overall effect size. If the measurement units differed, the standardized mean difference (SMD) was applied [47]. Heterogeneity among studies was evaluated using the I² statistic, categorized as low (25–50%), moderate (50–75%), and high (greater than 75%) [48]. A fixed-effects model was used when heterogeneity was less than 25%, and a random-effects model was applied when it exceeded 25% [49]. A significance level of 0.05 was employed for all statistical tests.

Meta-analyses were conducted on 18 variables related to lower limb landing mechanics to assess changes before and after fatigue intervention. These variables included:

1. Peak flexion angles of the knee, hip, and ankle joints.

2. Initial contact flexion angles of the hip and knee joints.

3. Peak abduction angles of the knee and hip joints and their angles at initial contact.

4. Hip external rotation angle at initial contact and peak internal rotation angle.

5. Peak internal adduction, internal rotation, extension, and flexion moments of the hip.

**Table 3. Quality scores of studies included in the review.**

| Study | N1 | N2 | N3 | N4 | N5 | N6 | N7 | N8 | N9 | N10 | Overall |
|---|---|---|---|---|---|---|---|---|---|---|---|
| Abbey C. Thomas et al. [36] | Y | NA | Y | NA | Y | Y | NA | NA | Y | Y | 6 |
| Anne Benjaminse et al. [37] | Y | NA | Y | Y | Y | Y | NA | NA | Y | Y | 7 |
| Danielle M.Brazen et al. [38] | Y | Y | Y | Y | Y | Y | NA | NA | Y | Y | 8 |
| DAVID R. BELL et al. [39] | Y | Y | Y | Y | Y | Y | NA | NA | Y | Y | 8 |
| Dominic Gehring et al. [11] | Y | NA | Y | NA | Y | Y | NA | NA | Y | Y | 6 |
| Evangelos Pappas et al. [40] | Y | NA | NA | NA | NA | Y | Y | NA | Y | Y | 5 |
| Kristı´n Briem et al. [41] | Y | NA | Y | Y | NA | Y | NA | NA | Y | Y | 6 |
| Lessi G.C et al. [42] | Y | Y | NA | Y | Y | Y | NA | NA | Y | Y | 7 |
| Marijeanne Liederbach et al. [21] | Y | NA | Y | Y | NA | Y | Y | NA | Y | Y | 7 |
| Michael P. Smith et al. [43] | Y | NA | Y | Y | Y | Y | NA | NA | Y | Y | 7 |
| Ram Haddas et al. [44] | Y | Y | NA | Y | Y | Y | Y | NA | Y | Y | 8 |
| Scott G. M et al. [45] | Y | Y | NA | Y | Y | Y | NA | NA | Y | Y | 7 |
| Thomas W. Kernozek et al. [12] | Y | Y | Y | Y | Y | Y | NA | NA | Y | Y | 8 |
| Zhang Qiang et al. [46] | Y | NA | NA | Y | NA | Y | NA | NA | Y | Y | 5 |

N = No, NA = Not Applicable, Y = Yes. The scale questions are in Table 2.

6. Peak extension and abduction moments of the knee joint.

7. Peak vertical ground reaction force (PVGRF).

To enhance the study's comparability and reliability, data included both males and females, and subgroup analyses were manually classified by gender. Each subgroup required a sufficient number of studies; subgroups with fewer than two studies were excluded from the final analysis.

## Results

Through electronic database searches, 11,057 records were identified. After the initial screening, 2,245 duplicate records were removed. Titles and abstracts of the remaining 8,812 records were screened, resulting in the exclusion of 8,657 records. A detailed evaluation of the 155 remaining full-text articles led to the exclusion of 136 based on study content and quality: 22 did not involve fatigue, 98 did not include both genders, 2 were non-English articles, 3 used 2D analysis methods, and 13 did not meet study requirements. After rigorous screening, 17 studies were included in the qualitative synthesis, and 14 were used in the quantitative synthesis. See S1 Fig in supplementary materials.

### Risk of bias in included articles

Table 2 details the quality scores of the included studies. Each column corresponds to a specific quality assessment criterion (N1 to N10), with each criterion scored out of 1 point, totaling a maximum of 10 points per study. The table lists the author names, their scores for each criterion, and the total score. Most studies scored between 5 and 8 out of 10, with an average score of 6.79. Nine out of the 14 studies were identified as "high quality," scoring 7 or higher.

### Characteristics of included studies

This meta-analysis included 14 studies involving different populations: six studies with healthy individuals, three with amateur athletes, two with collegiate athletes, and three with registered athletes, team athletes, and young individuals with a history of recurrent low back pain but currently asymptomatic. These studies encompassed a total of 496 participants, with 259 males and 237 females. Six studies used single-leg landing, while eight used double-leg landing. All fatigue induction protocols involved peripheral fatigue. The included studies provided extensive kinematic and kinetic results, covering peak flexion angles of the knee, hip, and ankle joints; initial contact flexion angles of the hip and knee joints; peak abduction angles of the knee and hip joints and their angles at initial contact; hip external rotation angle at initial contact and peak internal rotation angle; as well as peak internal adduction and internal rotation moments of the hip; peak extension and abduction moments of the knee joint; and peak vertical ground reaction force (PVGRF) (Table 4).

### Kinematic data of landing biomechanics

This meta-analysis included a total of 14 studies that examined the kinematic data of the knee, hip, and ankle joints, investigating the effects of fatigue and gender on these parameters. The kinematic data for each joint were represented by three-dimensional joint angles (in degrees). The specific parameters analyzed are detailed in Table 5.

### Knee kinematics

The meta-analysis focused on changes in knee flexion and abduction at initial contact, as well as peak flexion and abduction, before and after fatigue. See S2 Fig in supplementary materials. Key findings include:

1. Knee flexion angle at initial contact: Ten studies with 378 participants showed a total effect size of -0.72 (95% CI: -1.28, -0.17). Males exhibited a significant increase in angle (effect size: -1.23, 95% CI: -1.85, -0.61), while females showed an increasing trend that was not statistically significant (effect size: -0.25, 95% CI: -1.22, 0.72).

**Table 4. Demographic information from included studies.**

| Study | Training state | Male individu-als (n) | Female individu-als (n) | Age (years) | Fatigue protocol (Peripheral) | Landing test | outcomes |
|---|---|---|---|---|---|---|---|
| Abbey C. Thomas et al. 2010 [36] | Healthy people | 13 | 12 | 18-22 | 1. Subjects performed five alternating MVCCs, recording baseline peak torque 2. MVCC continued until torque dropped below 50% of baseline three times.A 20-second rest followed each drop. 3. This cycle repeated until torque fell below 50% during the initial five reps, indicating fatigue | Forward single-leg jump landing. | IC: hip/knee flexion angle; knee/hip IR angle; peak hip/knee flexion angle;peak hip/knee abduction angle;peak knee/hip IR angle;peak hip/knee flexion moment;peak hip/knee adduction moment;peak hip IR moment;peak knee ER moment; |
| Anne Benja-minse et al. 2008 [37] | healthy and physically active subjects | 15 | 15 | 18-30 | 1. Participants warmed up for 3 minutes, then ran at 5–8 mph with a 0% incline for another 3 minutes. 2. The incline increased by 2.5% every 2 minutes until exhaustion. 3. Verbal encouragement was provided throughout. 4. Average speeds were 6.3±0.62 mph for males and 5.8±0.43 mph for females. | Stop-jump task consisted of a single-leg | IC: knee flexion angle/ hip ER angle/hip adduction angle/Knee valgus–varus angles;peak knee valgus–varus; peak knee flexion angle; peak knee abduction angle;peak hip adduction angle;peak hip IR angle; |
| Danielle M. Brazen et al. 2010 [38] | Healthy people | 12 | 12 | 19.5±3.8 | Repeat the following agility training circuit five times: 1. Ladder Agility Drills. 2. Side-to-Side Jumps. 3. Mini Trampoline Jumps. 4. Mini Hurdle Jumps. 5. V-shaped Vertical Jumps. | Single-leg drop landing from a 0.36m height. | Peake VGRF |
| DAVID R. BELL et al. 2016 [39] | Healthy people | 20 | 20 | 18-23 | 1. Participants start at the first cone, spaced 4.05 meters apart. 2. After completing the circuit, they perform a 30-second wall sit with knees bent at 90 degrees and back against the wall. 3. Then, they execute 10 rapid double-leg vertical jumps. 4. Finally, they hold a 30-second plank on a 5mm yoga mat | Double-leg drop jump from a 30 cm height. | Peak VGRF |
| Dominic Gehring et al. 2009 [11] | Healthy people | 13 | 13 | 24±2.4 | 1. Participants used the BERI MED leg press machine and flexed/extended their knees to a 90% angle. 2. Leg contractions were performed at 50% of the maximum load, with a maximum of 1 repetition. 3. The experiment continued until partici-pants couldn't complete the load. | Double-leg drop landing from a 52 cm height. | IC knee flexion angle;peak knee flexion angle; IC knee abduction angle;peak knee abduction angle;Knee flexion velocity 0–50 ms (°/s); Knee flexion velocity 50–100 ms (°/s) Peak VGRF |
| Evangelos Pappas et al. 2009 [40] | Recre-ational athletes | 14 | 15 | 20-40 | 1. Jumping over obstacles: Participants cleared a series of five consecutive obsta-cles measuring 5–7 centimeters in height. 2. Repetitions: The jumping sequence was repeated 20 times. 3.Total jumps: The participants completed a total of 100 jumps | Double-leg drop landing from a 40 cm height. | Peak knee flexion angle |

*(Continued)*

| Study | Training state | Male individuals (n) | Female individuals (n) | Age (years) | Fatigue protocol (Peripheral) | Landing test | outcomes |
|---|---|---|---|---|---|---|---|
| Kristı́n Briem et al. 2017 [41] | Registered athletes | 68 | 48 | mean age,10.4 | 1. Starting position: Participants started at both ends, with bumper bars positioned at a distance of 1.5 times their leg length.<br>2. Lateral push and slide: Participants laterally pushed off from one bumper bar and slid to the opposite side of the skateboard.<br>3. Duration: The activity was repeated for 5 minutes.<br>4. Endurance: Participants continued the activity until they could no longer sustain it | Double-leg drop jump from a 25 cm height. | IC knee flexion angle;peak knee flexion angle;<br>Peak VGRF |
| Lessi G.C et al. 2017 [42] | Recreational athletes | 20 | 20 | 18-30 | 1. 10 sets of bilateral squats (90° knee flexion).<br>2. 2 sets of bilateral maximum effort vertical jumps and 20 steps (31 cm high stairs). Participants use their dominant leg for stepping | Single-leg drop vertical jump from a 31 cm height | IC: Knee sagittal plane; Knee frontal plane; Hip sagittal plane; Hip frontal plane; Peak: Knee sagittal plane; Knee frontal plane; Hip sagittal plane; Hip frontal plane; Flexion (+); extension (-); abduction (+); adduction (-) |
| Mari-jeanne Liederbach et al. 2014 [21] | Team sport athletes | 20 | 20 | 20-22 | 1. Climbing 50 steps on a 30 cm box.<br>2. Performing 15 maximum effort single-leg vertical jumps | Single-leg drop landing from a 30 cm height. | Peak knee abduction angle; Knee abduction angle at IC; Peak knee abduction moment; Peak hip adduction angle; Hip adduction angle at IC;<br>Peak hip adduction moment; Peak hip IR angle; Hip IR angle at IC; Peak hip IR moment; Knee flexion angle at IC; Peak knee flexion angle; Peak knee flexion moment |
| Michael P. Smith et al. 2009 [43] | healthy and physically active subjects | 12 | 14 | 18-35 | Participants performed knee flexion to 60 degrees. They repeated 15-second maximum isometric contractions followed by 5 seconds of relaxation until reaching fatigue (below 50% MVC) | Double-leg drop landing from a 50 cm height. | Peak VGRF; IC knee flexion angle; |
| Ram Haddas et al. 2015 [44] | Young adults without current symptoms but with recurrent LBP | 17 | 15 | 21.65±2.30 | Participants fatigued by performing submaximal free weight squats using 15% of their body weight until task failure. | Double-leg drop vertical jump from a 30 cm height | Knee sagittal-plane angle at IC;Knee frontal-plane angle at IC;Knee transverse-plane (rotation) angle at IC; Peak sagittal-plane knee angle;Peakfrontal-plane knee angle;Peak knee transverse-plane (rotation) angle; Peak VGRF;Peak: Knee sagittal-plane moment/Knee frontal-plane moment/Ankle sagittal-plane moment/Ankle frontal-plane moment The convention for positive angles is flexion, abduction, and external rotation. |

*(Continued)*

**Table 4.** (Continued)

| Study | Training state | Male individuals (n) | Female individuals (n) | Age (years) | Fatigue protocol (Peripheral) | Landing test | outcomes |
|---|---|---|---|---|---|---|---|
| Scott G. M et al. 2007 [45] | College athlete | 10 | 10 | 20.7 ± 1.3 | Participants performed explosive jumps on a 20 cm high step, covering a 6-meter distance with a change in direction. They landed in a deep knee flexion position and quickly rebounded to the next jump. The process was repeated as many times as possible within 4 minutes. | Double-leg drop vertical jump from a 50 cm height | IC Joint: Hip flexion/Hip abduction/Hip ER/Knee flexion/Knee adduction/Knee ER/Ankle dorsiflexion/Ankle IR;Peak Joint:Hip flexion/Hip adductionn/Hip IR/Knee flexion/Knee abduction/Knee IR/Ankle dorsiflexion/Ankle ER; Peak Moment:Hip flexion/Hip adduction/Hip IR/Knee flexion/Knee abduction/Knee IR/Ankle dorsiflexion/Ankle ER |
| Thomas W. Kernozek et al. 2008 [12] | Recreational athletes | 16 | 14 | 23.0 ± 0.9 | Participants performed as many repetitions as possible with a set weight, controlling the speed of each phase. Resting 90 seconds between sets, they continued until unable to lift the weight. | Single-leg drop landing from a 50 cm height. | IC, peak joint: Hip abductor/Hip flexion/Knee flexion;Peakjoint Knee abduction/Ankle dorsiflexion; Peak:Knee abduction moment/Knee extension moment;PeakVGRF |
| Zhang Qiang et al. 2021 [46] | College athlete | 9 | 9 | 21.3 ± 1.0 | Participants stood at the center of a 6-meter diameter circular area with six evenly spaced lights around it. A controller randomly lit a series of lights, and participants had to quickly move to touch and turn off the active light, triggering the illumination of the next light. | Double-leg drop landing from a 40 cm height. | IC:Ankle plantarflexion angle/Knee flexion angle/Knee add/abduction angle/Hip flexion angle; Peak:ankle dorsiflexion angle/knee flexion angle/Peak knee abduction angle/hip flexion angle/hip abduction angle; Peak Moment:ankle plantarflexion/knee extension/knee adduction/hip extension/hip abduction |

MVCC: maximum voluntary concentric contractions; IC: initial contact; IR: internal rotation; ER: external rotation; VGRF: peak vertical ground reaction force.

2. Peak knee flexion angle: Fatigue did not significantly affect peak flexion angle for both males and females (total effect size: -0.96, 95% CI: -2.93, 1.00; males: -1.42, 95% CI: -5.19, 2.34; females: -0.75, 95% CI: -3.13, 1.63).

3. Knee abduction angle at initial contact: The total effect size was -0.26 (95% CI: -0.55, 0.04), with both genders showing a non-significant increasing trend post-fatigue (males: -0.13, 95% CI: -0.57, 0.31; females: -0.36, 95% CI: -0.76, 0.04).

4. Peak knee abduction angle: Changes in peak abduction angle were not statistically significant for both genders (total effect size: 0.23, 95% CI: -0.13, 0.60; males: 0.41, 95% CI: -0.12, 0.95; females: 0.07, 95% CI: -0.44, 0.58).

**Hip kinematics**

The meta-analysis of hip biomechanics before and after fatigue covered six outcome measures: hip flexion and abduction at initial contact, peak flexion and abduction, and external and internal rotation movements. See S3 Fig in supplementary materials. Key findings include:

**Table 5. Summary of a meta-analysis evaluating the effects of fatigue before and after (including gender subgroups) and on major biomechanical variables of lower limb landing mechanics.**

| Kinematic variables | Sub-group | No. of studies | N | Subtotal I² | Z | P | Subtotal (95% CI) | Total I² | Z | P | Total (95% CI) | Effect measure/ Analysis model |
|---|---|---|---|---|---|---|---|---|---|---|---|---|
| Knee flexion at IC | Male | 10 | 200 | 0% | 3.9 | <0.001* | -1.23 [-1.85, -0.61] | 16% | 2.55 | 0.01* | -0.72 [-1.28, -0.17] | MD/Random |
| | Female | 10 | 178 | 32% | 0.51 | 0.61 | -0.25 [-1.22, 0.72] | | | | | |
| Peak knee flexion | Male | 10 | 147 | 84% | 0.74 | 0.46 | -1.42 [-5.19, 2.34] | 77% | 0.96 | 0.34 | -0.96 [-2.93, 1.00] | MD/Random |
| | Female | 10 | 143 | 64% | 0.62 | 0.54 | -0.75 [-3.13, 1.63] | | | | | |
| Knee abduction at IC | Male | 4 | 66 | 0% | 0.58 | 0.56 | -0.13 [-0.57, 0.31] | 0% | 1.71 | 0.09 | -0.26 [-0.55, 0.04] | MD/Fixed |
| | Female | 6 | 89 | 0% | 1.78 | 0.07 | -0.36 [-0.76, 0.04] | | | | | |
| Peak knee abduction | Male | 9 | 135 | 0% | 1.52 | 0.13 | 0.41 [-0.12, 0.95] | 0% | 1.25 | 0.21 | 0.23 [-0.13, 0.60] | MD/Fixed |
| | Female | 8 | 118 | 0% | 0.27 | 0.79 | 0.07 [-0.44, 0.58] | | | | | |
| Hip flexion at IC | Male | 5 | 68 | 0% | 0.38 | 0.7 | 0.48 [-1.98, 2.93] | 0% | 0.21 | 0.83 | 0.17 [-1.44, 1.79] | MD/Fixed |
| | Female | 5 | 65 | 0% | 0.05 | 0.96 | -0.06 [-2.20, 2.09] | | | | | |
| Peak Hip flexion | Male | 5 | 68 | 0% | 0.36 | 0.72 | 0.63 [-2.83, 4.09] | 0% | 0.39 | 0.7 | -0.53 [-3.20, 2.14] | MD/Fixed |
| | Female | 5 | 65 | 0% | 1.04 | 0.3 | -2.21 [-6.39, 1.97] | | | | | |
| Hip abduction at IC | Male | 5 | 74 | 0% | 0.16 | 0.87 | -0.13 [-1.70, 1.44] | 0% | 0.37 | 0.71 | -0.18 [-1.14, 0.77] | MD/Fixed |
| | Female | 5 | 71 | 0% | 0.35 | 0.73 | -0.22 [-1.43, 0.99] | | | | | |
| Peak Hip abduction | Male | 4 | 53 | 0% | 0.19 | 0.85 | -0.19 [-2.09, 1.72] | 1% | 0.2 | 0.84 | -0.14 [-1.48, 1.20] | MD/Random |
| | Female | 4 | 50 | 47% | 0.21 | 0.83 | 0.29 [-2.36, 2.94] | | | | | |
| Hip ER at IC | Male | 3 | 45 | 0% | 3.17 | 0.002* | 1.35 [0.51, 2.18] | 0% | 4.31 | <0.001* | 1.27 [0.69, 1.85] | MD/Fixed |
| | Female | 3 | 45 | 0% | 2.93 | 0.003* | 1.20 [0.40, 2.01] | | | | | |
| Peak Hip IR | Male | 4 | 58 | 51% | 0.98 | 0.33 | -1.26 [-3.77, 1.25] | 41% | 1.15 | 0.25 | -1.10 [-2.96, 0.77] | MD/Random |
| | Female | 3 | 37 | 0% | 0.03 | 0.97 | -0.05 [-2.79, 2.70] | | | | | |
| Peak Ankle dorsiflexion | Male | 3 | 35 | 0% | 1.04 | 0.3 | -1.61 [-4.64, 1.41] | 0% | 1.7 | 0.09 | -1.69 [-3.64, 0.26] | MD/Fixed |
| | Female | 3 | 33 | 0% | 1.35 | 0.18 | -1.75 [-4.29, 0.79] | | | | | |
| Kinetics variables | | | | | | | | | | | | |
| Peak Hip IR moment | Male | 3 | 43 | 61% | 0.97 | 0.33 | -0.35 [-1.05, 0.35] | 43% | 1.95 | 0.05 | -0.41 [-0.83, 0.00] | SMD/Random |
| | Female | 3 | 42 | 45% | 1.48 | 0.14 | -0.45 [-1.06, 0.15] | | | | | |

*(Continued)*

**Table 5.** (Continued)

| Kinematic variables | Sub-group | No. of studies | N | Subtotal | | | Subtotal (95% CI) | Total | | | Total (95% CI) | Effect measure/ Analysis model |
|---|---|---|---|---|---|---|---|---|---|---|---|---|
| | | | | I² | Z | P | | I² | Z | P | | |
| Peak Hip flexion moment | Male | 2 | 23 | 26% | 0.83 | 0.41 | 0.29 [-0.40, 0.98] | 0% | 1.57 | 0.12 | 0.34 [-0.08, 0.76] | SMD/Random |
| | Female | 2 | 22 | 18% | 1.18 | 0.24 | 0.40 [-0.26, 1.07] | | | | | |
| Peak hip extension moment | Male | 2 | 25 | 0% | 2.01 | 0.04* | 0.58 [0.01, 1.15] | 0% | 2.43 | 0.01* | 0.51 [0.10, 0.92] | SMD/Fixed |
| | Female | 2 | 23 | 0% | 1.43 | 0.15 | 0.43 [-0.16, 1.01] | | | | | |
| Peak hip adduction moment | Male | 3 | 43 | 65% | 0.8 | 0.43 | 0.30 [-0.44, 1.05] | 44% | 1.59 | 0.11 | 0.34 [-0.08, 0.75] | SMD/Random |
| | Female | 3 | 43 | 37% | 1.23 | 0.22 | 0.35 [-0.21, 0.91] | | | | | |
| Peak knee extension moment | Male | 4 | 55 | 29% | 3.15 | 0.002* | 0.76 [0.29, 1.23] | 4% | 4.07 | <0.001* | 0.64 [0.33, 0.95] | SMD/Random |
| | Female | 3 | 36 | 0% | 1.99 | 0.05 | 0.48 [0.01, 0.95] | | | | | |
| Peak Knee abduction moment | Male | 4 | 59 | 51% | 0.26 | 0.79 | -0.07 [-0.60, 0.46] | 73% | 0.71 | 0.48 | -0.19 [-0.72, 0.34] | SMD/Random |
| | Female | 4 | 56 | 85% | 0.65 | 0.52 | -0.34 [-1.39, 0.70] | | | | | |
| Peak VGRF | Male | 7 | 158 | 63% | 0.09 | 0.93 | 0.02 [-0.39, 0.42] | 54% | 0.43 | 0.67 | 0.06 [-0.20, 0.31] | SMD/Random |
| | Female | 7 | 136 | 49% | 0.53 | 0.59 | 0.10 [-0.26, 0.45] | | | | | |

Most forest plots are in Appendix D. IC: initial contact; IR: internal rotation; ER: external rotation; VGRF: peak vertical ground reaction force; MD: mean difference; SMD: standardization mean difference; Moment (Nm/kg or N m/kg·m or N·m/kg of BW*H or N/kg). Peak vertical ground-reaction force (BW or %BW or N or N/kg). BW, body weight; H, height; *: P<0.05.

1. Hip flexion angle at initial contact: Males (5 studies, N=68) had an effect size of 0.48 (95% CI: -1.98, 2.93), and females (5 studies, N=65) had an effect size of -0.06 (95% CI: -2.20, 2.09), showing no significant changes.

2. Peak hip flexion angle: Males (5 studies, N=68) had an effect size of 0.63 (95% CI: -2.83, 4.09), and females (5 studies, N=65) had an effect size of -2.21 (95% CI: -6.39, 1.97), showing no significant changes, though females trended towards increased angles post-fatigue.

3. Hip abduction angle at initial contact: Males (5 studies, N=74) had an effect size of -0.13 (95% CI: -1.70, 1.44), and females (5 studies, N=71) had an effect size of -0.22 (95% CI: -1.43, 0.99), showing no significant changes.

4. Peak hip abduction angle: Males (4 studies, N=53) had an effect size of -0.19 (95% CI: -2.09, 1.72), and females (4 studies, N=50) had an effect size of 0.29 (95% CI: -2.36, 2.94), showing no significant changes.

5. Hip external rotation angle at initial contact: Both genders showed significant positive changes, with males (3 studies, N=45) having an effect size of 1.35 (95% CI: 0.51, 2.18) and females (3 studies, N=45) having an effect size of 1.20 (95% CI: 0.40, 2.01), indicating a trend of angle reduction post-fatigue.

6. Peak hip internal rotation angle: Males (4 studies, N=58) had an effect size of -1.26 (95% CI: -3.77, 1.25), and females (3 studies, N=37) had an effect size of -0.05 (95% CI: -2.79, 2.70), showing no significant changes, although males trended towards increased angles.

Overall, while some changes were observed, most results regarding hip flexion and abduction were not statistically significant, with only hip external rotation at initial contact showing significant changes in both genders.

## Ankle kinematics

In this meta -analysis, changes in peak ankle dorsiflexion angle before and after fatigue were evaluated across genders. The overall effect size was -1.69 (95% CI: -3.64, 0.26). For males (3 studies, N = 35), the effect size was -1.61 (95% CI: -4.64, 1.41), and for females (3 studies, N = 33), it was -1.75 (95% CI: -4.29, 0.79), both showing no significant changes, and heterogeneity (I²) was 0% for both. These results suggest that fatigue may lead to an increase in ankle dorsiflexion angle and indicate no significant difference in changes between males and females under fatigue, with the changes not significantly influenced by sample heterogeneity. See S4 Fig in supplementary materials.

## Kinetics data of landing biomechanics

The dynamic data for the hip and knee joints, as well as peak vertical ground reaction force (PVGRF), were evaluated.

## Knee kinetics

This meta-analysis examined the effects of fatigue on knee extension and abduction torques, considering gender differences. For Peak knee extension moment, males (N = 55) exhibited an effect size of 0.76 (95% CI: 0.29, 1.23, P = 0.002), indicating a significant change, while females (N = 36) had an effect size of 0.48 (95% CI: 0.01, 0.95, P = 0.05), which was nearly significant. The overall effect size was 0.64 (95% CI: 0.33, 0.95, P < 0.001), demonstrating a significant change. Heterogeneity was 29% for males, 0% for females, and 4% overall, indicating a significant reduction in knee extension torque due to fatigue.

For Peak Knee abduction moment, males (N = 59) showed an effect size of -0.07 (95% CI: -0.60, 0.46, P = 0.79), and females (N = 56) had an effect size of -0.34 (95% CI: -1.39, 0.70, P = 0.52). The overall effect size was -0.19 (95% CI: -0.72, 0.34, P = 0.48), with none showing significant changes. Heterogeneity was 51% for males, 85% for females, and 73% overall. These results suggest a statistically significant effect of fatigue on knee extension moment in males, but not on abduction torque for either gender, with high heterogeneity observed. See S5 Fig in supplementary materials.

## Hip kinetics

This meta-analysis assessed changes in hip internal rotation, flexion, extension, and adduction torques before and after fatigue across genders. For peak hip internal rotation moment, males (3 studies, N = 43) had an effect size of -0.35 (95% CI: -1.05, 0.35, P = 0.33), and females (3 studies, N = 42) had -0.45 (95% CI: -1.06, 0.15, P = 0.14). The overall effect size was -0.41 (95% CI: -0.83, 0.00, P = 0.05), with no significant changes. For peak hip flexion moment, males (2 studies, N = 23) had an effect size of 0.29 (95% CI: -0.40, 0.98, P = 0.41), and females (2 studies, N = 22) had 0.40 (95% CI: -0.26, 1.07, P = 0.24). The overall effect size was 0.34 (95% CI: -0.08, 0.76, P = 0.12), with no significant changes. For peak hip extension moment, males (2 studies, N = 25) had an effect size of 0.58 (95% CI: 0.01, 1.15, P = 0.04), and females (2 studies, N = 23) had 0.43 (95% CI: -0.16, 1.01, P = 0.15). The overall effect size was 0.51 (95% CI: 0.10, 0.92, P = 0.01), showing significant changes in males and the overall analysis, indicating a decrease. For peak hip adduction moment, males (3 studies, N = 43) had an effect size of 0.30 (95% CI: -0.44, 1.05, P = 0.43), and females (3 studies, N = 43) had 0.35 (95% CI: -0.21, 0.91, P = 0.22). The overall effect size was 0.34 (95% CI: -0.08, 0.75, P = 0.11), with no significant changes. These findings highlight the potential impact of fatigue on different hip joint moment, with significant changes particularly in male hip extension moment. See S6 Fig in supplementary materials.

**Peak VGRF**

The meta-analysis of peak vertical ground reaction force (PVGRF) examined gender differences in biomechanical responses to fatigue. The effect size for males (7 studies, N = 158) was 0.02 (95% CI: -0.39, 0.42, P = 0.93), and for females (7 studies, N = 136), it was 0.10 (95% CI: -0.26, 0.45, P = 0.59), neither indicating significant changes. The overall effect size was 0.06 (95% CI: -0.20, 0.31, P = 0.67), also showing no significant changes. Heterogeneity was 63% for males, 49% for females, and 54% overall. These results indicate no significant differences in peak vertical ground reaction force between genders before and after fatigue, with moderate heterogeneity. The findings suggest that fatigue does not significantly impact peak vertical ground reaction force across genders, and variability among study samples may affect result consistency. See S7 Fig in supplementary materials.

## Discussion

This meta-analysis systematically reviewed existing literature to understand how fatigue influences gender differences in landing biomechanics and the associated risk of ACL injuries. The results indicate that fatigue significantly affects the kinematics and kinetics of the knee, hip, and ankle joints. However, most gender differences in these effects were not statistically significant, with only a few variables showing notable changes. This discussion will delve into our key findings, focusing on the landing biomechanics of the knee, hip, and ankle, while offering a deeper analysis of the potential mechanisms behind observed gender differences.

In the sagittal plane biomechanics, our study found that males had a greater knee flexion angle at initial contact compared to females, both before and after fatigue. Under fatigue, both genders showed an increase in knee flexion angle, but the increase was more significant in males.This may be attributed to the greater strength and control over the knee joint in males, which allows for more noticeable alterations in landing biomechanics under fatigue, as they depend on neuromuscular adjustments to stabilize the knee during landing. In contrast, females may rely more on passive structures, such as ligaments and joint flexibility, leading to less pronounced changes under fatigue [50,51]. For peak knee flexion angles, both genders showed an increasing trend post-fatigue, which is considered a compensatory mechanism after fatigue and also a strategy to reduce ACL loading [52]. Decker et al.'s study supports this perspective but additionally observed that, in a non-fatigued state, females exhibit greater knee flexion angles during landing compared to males [8]. This gender difference may be attributed to females' higher flexibility during physical activity [53] and distinct muscle strength distributions [54], both of which influence their landing mechanics and potential injury risks. Our study found that males typically exhibit smaller peak knee flexion angles before fatigue compared to females, which may be attributed to their greater muscle strength and mass, both of which enhance knee stability and facilitate earlier stabilization. The greater musculature in males enables more effective control of knee movement, thereby reducing excessive knee flexion at initial contact [55,56]. However, under fatigue, this greater muscle strength facilitates a more pronounced compensatory increase in knee flexion, as males rely more on muscular control to stabilize the knee during landing.

The importance of sagittal plane knee movements lies in their relationship with anterior-posterior knee shear forces. There is considerable debate on whether abnormal sagittal plane dynamics can cause anterior cruciate ligament (ACL) injuries. Chappell et al. [22] noted that increased anterior tibial shear forces during stop-jumping were associated with increased knee flexion angles. Our study found that the reduction in knee extension torque post-fatigue was more significant in males, with an effect size of 0.76, compared to 0.48 in females. This may be due to the downward momentum reduction caused by knee flexion during landing, along with eccentric contraction of the surrounding knee muscles [8]. The human body's inherent ability to adapt and regulate enables it to modify lower limb kinematics, thereby reducing the risk of ACL injuries [57]. This can be considered a natural protective mechanism against landing impacts. However, while knee flexion can help mitigate impact forces to some extent, improper flexion may compromise the stability of the medial collateral ligament (MCL) and lateral collateral ligament (LCL), resulting in rotational instability of the knee and an increased risk of ACL injuries [25]. This observation further elucidates why ACL injuries frequently occur during the landing

phase of basketball [58]. However, a study by Schmitz et al. [59] on 90-minute intermittent exercise found that as exercise duration increased, subjects landed in a more upright posture post-fatigue, with less hip and knee flexion. Reduced joint flexion was accompanied by decreased hip work and increased knee shear forces [60]. Some studies noted that compared to males, females showed increased quadriceps activation before landing in the same tasks [61]. Krosshaug et al. [62] found that female basketball players exhibited significantly greater knee flexion angles at initial ground contact and within the first 50 milliseconds after contact compared to males, suggesting that females might use different biomechanical strategies during landing to better absorb impact forces. Increased quadriceps activation during landing preparation might increase ACL loading [63]. However, this activation also plays a critical role in reducing rotational stress on the knee joint. In fact, the protective effect of quadriceps activation against rotational instability can, to some extent, offset its contribution to increased ACL tension [64,65]. Regarding hip extension (2 studies) and flexion (2 studies) torques, as well as peak ankle dorsiflexion angles (3 studies), despite being low risk, the limited number of studies reporting these indicators prevents drawing definitive conclusions on potential biomechanical differences in landing mechanics between genders under fatigue.

Frontal plane biomechanics are also related to ACL injuries. Our study found no significant differences in knee and hip abduction angles at initial contact and peak values. However, under fatigue, females showed a greater trend of increased knee abduction angles compared to males. Evidence suggests that excessive dynamic knee valgus may increase ACL tear risk [7,66]. Hewett et al. [4] conducted a prospective study on female athletes in high-risk sports (such as soccer, basketball, and volleyball) during jumping and landing tasks. Using three-dimensional motion capture, they measured joint angles to assess neuromuscular control and kinetic methods to measure joint loads (joint torques). Among the participants, 9 athletes sustained ACL injuries. The ACL injury group had a knee abduction angle of 8° during landing, significantly higher than the non-injured group. Additionally, their knee abduction torque was 2.5 times greater, and the ground reaction force was 20% higher. The injured athletes also showed a faster increase in angular velocity, ground reaction force, and joint torque compared to non-injured athletes. Chappell and Yu et al. [67] further showed that during jumping and landing tasks, female athletes exhibited greater valgus torques compared to males. It is worth noting that compared to males, females generally have a larger Q angle, which refers to the relationship of the femoral axis relative to the tibial axis in the frontal plane of the knee joint and is often associated with increased knee valgus [68,69]. However, no significant relationship has been found between the Q angle and ACL injury rates in female athletes [70,71].

Horizontal plane biomechanics primarily focus on the movement mechanisms of the hip and knee joints. However, this study only included some hip joint mechanical indicators. We found that under fatigue, the hip external rotation angle at initial contact significantly decreased in both male and female athletes. This change could increase the load on the knee and ankle joints, thereby raising the risk of ACL injuries. Koga et al.'s study [72] using a 3D MBIM system found significant differences in hip joint positions between ACL injury cases during actual competitions and non-injury conditions, as well as during side-cutting and landing movements. In injury cases, the hip joint typically showed a smooth flexion transition after ground contact, whereas ACL injuries were characterized by significant hip internal rotation, causing internal rotation of the knee and ankle joints. This internal rotation increased the load on the knee joint, raising the risk of ACL injuries.

Additionally, Koga et al.'s research [72] indicated that during side-cutting and landing movements, hip internal rotation affects not only the hip joint itself but also negatively impacts the entire lower limb kinetic chain, increasing stress and injury risks to other joints. This suggests that future research on ACL injury mechanisms should focus more on the dynamic performance of the hip joint and its coordination with other lower limb joints.

## Conclusion

This study employed a systematic review and meta-analysis to investigate the effects of fatigue and gender differences on lower limb landing biomechanics and their relationship with ACL injury risk. While most results did not show significant changes, some important trends and significant findings were identified.

First, fatigue significantly increased the knee flexion angle at initial contact in males, which helps reduce ACL loading. Second, males showed a significant reduction in knee extension torque after fatigue, potentially affecting knee stability and increasing injury risk. Additionally, fatigue significantly decreased the hip external rotation angle at initial contact in both males and females, which may increase the load on the knee and ankle joints, raising the risk of injury. However, other hip torques (internal rotation, flexion, and adduction) did not show significant changes under fatigue. Finally, the ankle dorsiflexion angle and peak vertical ground reaction force (VGRF) did not show significant changes before and after fatigue, indicating limited effects of fatigue on these parameters.

Overall, fatigue significantly affects certain knee and hip joint parameters, especially in males. These findings underscore the importance of considering gender differences in fatigue management and sports strategies to reduce ACL injury risk. Future research should further explore the mechanisms behind these changes and assess gender-specific interventions to mitigate ACL injury risk.

## Limitations

The study has several limitations that should be considered. First, although 14 studies were included, only a few reported ankle biomechanical variables, which limited the scope of the analysis. This restriction hinders the investigation of the impact of ankle biomechanics on lower limb kinematics and ACL injury risk during landing. Second, the review focused solely on lower limb biomechanical indicators, without considering the positioning and control of the trunk and the entire kinetic chain, which could significantly influence knee biomechanics [73]. Neglecting potential interactions across the kinetic chain, including trunk and hip control, may lead to an incomplete understanding of knee biomechanics. This limitation may underestimate the interdependencies between body segments and their effect on ACL injury mechanisms. Third, while the study emphasized gender, it did not perform a subgroup analysis of the effects of different landing styles and fatigue protocols on lower limb biomechanics. Variations in landing styles and fatigue protocols could alter the biomechanics of lower limb joints during landing. Finally, although the 14 studies included in this review reported data on both sexes before and after fatigue, studies focusing on a single sex were excluded. This omission may affect the comprehensive understanding of gender differences. Future research should address these limitations by incorporating a broader range of biomechanical variables, conducting subgroup analyses, and including sex-specific studies to provide a more comprehensive understanding of ACL injury mechanisms and prevention strategies.

## Supporting information

**S1 Fig. PRISMA flow chart for inclusion and exclusion of studies.**
(TIF)

**S2 Fig. The forest plot of the effects of fatigue on knee joint kinematics and gender differences.**
(TIF)

**S3 Fig. The forest plot of the effects of fatigue on hip joint kinematics and gender differences.**
(TIF)

**S4 Fig. The forest plot of the effects of fatigue on ankle joint kinematics and gender differences.**
(TIF)

**S5 Fig. The forest plot of the effects of fatigue on knee joint kinetics and gender differences.**
(TIF)

**S6 Fig. The forest plot of the effects of fatigue on hip joint kinetics and gender differences.**
(TIF)

**S7 Fig. The forest plot of the effects of fatigue on vertical ground reaction force and gender differences.**
(TIF)

**S1 File. PRISMA_2020_checklist.**
(DOCX)

**S2 File. Data availability.**
(DOCX)

## Acknowledgments

Thank you to all the researchers who have contributed to this study.

## Author contributions

**Conceptualization:** Chengxun Liu, Jiaxin He, Bojin Cheng.

**Data curation:** Chengxun Liu, Wuwen Peng, Wenhao Qu, Zhiyong Zhang, Duanying Li.

**Formal analysis:** Wenhao Qu, Duanying Li.

**Funding acquisition:** Jian Sun.

**Investigation:** Wuwen Peng, Zhiyong Zhang, Bojin Cheng, Duanying Li.

**Methodology:** Wuwen Peng, Zhiyong Zhang, Bojin Cheng, Duanying Li.

**Project administration:** Jian Sun, Jiaxin He, Bojin Cheng, Duanying Li.

**Resources:** Jiaxin He, Duanying Li.

**Software:** Wenhao Qu, Zhiyong Zhang, Bojin Cheng.

**Supervision:** Chengxun Liu, Jiaxin He, Bojin Cheng.

**Validation:** Chengxun Liu, Jiaxin He, Bojin Cheng.

**Visualization:** Chengxun Liu, Wenhao Qu, Zhiyong Zhang, Bojin Cheng, Duanying Li.

**Writing – original draft:** Chengxun Liu.

**Writing – review & editing:** Chengxun Liu, Bojin Cheng.

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
