## [Decision Letter · Decision Letter 0]

23 Dec 2024

PONE-D-24-38651Gender Differences in the Impact of Fatigue on Lower Limb Landing Biomechanics and Their Association with Anterior Cruciate Ligament (ACL) Injuries: A Sysstematic Review and Meta-AnalysisPLOS ONE

Dear Dr. Cheng,

Thank you for submitting your manuscript to PLOS ONE. After careful consideration, we feel that it has merit but does not fully meet PLOS ONE’s publication criteria as it currently stands. Therefore, we invite you to submit a revised version of the manuscript that addresses the points raised during the review process.

 The paper has been revised by to experts in the filed, and both of them were positive about the work. However, they highlighted a number of points that should be carefully addressed, in order to improve the overall quality of the manuscript. Presentation of some aspects, and some methodological choices should also be clarified.

We look forward to receiving your revised manuscript.

Kind regards,

Alessandro Mengarelli

Academic Editor

PLOS ONE

Journal Requirements:

2. Please ensure that your PRISMA flow diagram is included in your main manuscript file as Figure 1 which is now in Figure 2; please see the PLOS ONE submission guidelines for systematic reviews and meta-analyses at https://journals.plos.org/plosone/s/submission-guidelines#loc-systematic-reviews-and-meta-analyses .

Please confirm at this time whether or not your submission contains all raw data required to replicate the results of your study. Authors must share the “minimal data set” for their submission. PLOS defines the minimal data set to consist of the data required to replicate all study findings reported in the article, as well as related metadata and methods (https://journals.plos.org/plosone/s/data-availability#loc-minimal-data-set-definition ).

If your submission does not contain these data, please either upload them as Supporting Information files or deposit them to a stable, public repository and provide us with the relevant URLs, DOIs, or accession numbers. For a list of recommended repositories, please see https://journals.plos.org/plosone/s/recommended-repositories .

6. Please amend either the title on the online submission form (via Edit Submission) or the title in the manuscript so that they are identical.

7. Please upload a copy of Supporting Information Figure/Table/etc. S1 to S11 which you refer to in your text on pages 44 and 45.

8. As required by our policy on Data Availability, please ensure your manuscript or supplementary information includes the following:

Reviewers' comments:

Reviewer's Responses to Questions

**Comments to the Author**

1. Is the manuscript technically sound, and do the data support the conclusions?

Reviewer #1: Yes

Reviewer #2: Partly

2. Has the statistical analysis been performed appropriately and rigorously? 

Reviewer #1: Yes

Reviewer #2: Yes

3. Have the authors made all data underlying the findings in their manuscript fully available?

Reviewer #1: Yes

Reviewer #2: Yes

4. Is the manuscript presented in an intelligible fashion and written in standard English?

Reviewer #1: Yes

Reviewer #2: Yes

5. Review Comments to the Author

Reviewer #1: Thank you for giving me the opportunity to review this valuable study. This research is a systematic review and meta-analysis focusing on lower limb biomechanics before and after fatigue tasks, and it is very well designed. The results are clear and reasonable, and I believe a minor revision is appropriate. Below are my comments:

• Line 98-99: Please describe the method used to ensure agreement among investigators (e.g., whether recommendations by grade were applied).

• Line 173: Yes/No responses for items 1-10 in Table 2 are missing.

• Line 181: Would it be more appropriate to use "peripheral" instead of "isolated"? Similarly, consider whether "central" would be better than "whole body."

• Table 3: It may improve readability if you classify the fatigue protocol as either central or peripheral before providing detailed protocol information.

• Line 213-214: This content should be in the Discussion section, not in the Results section.

• Line 319-320: The phrase "strategy to reduce ACL loading" may be better interpreted as a compensatory mechanism in response to fatigue, rather than simply reducing the load on the ACL.

• Line 330-334: If this study does not compare athletes with non-athletes, this statement is unnecessary. Furthermore, concluding that individual responses vary could undermine the overall meaning of this review, so caution is advised.

• Line 342: As mentioned earlier, this is not about reducing the risk of ACL injuries through fatigue-induced behavior, but rather a compensatory movement in response to fatigued muscles. It's also important to note that slight knee flexion does not necessarily increase the risk of ACL injury; in fact, mild flexion could relax the MCL and LCL, potentially contributing to rotational instability. Please revisit this point from the Introduction onward.

• Line 350-351: Reference 58 reflects an in vivo perspective, and while increased quadriceps contraction can increase tension on the ACL, it also suppresses rotational stress. Please take this fact into account in your discussion.

• Line 368-375: While the content is not incorrect, if this is not a review focusing on electromyography, this discussion may be unnecessary.

• Line 416: Please add a limitation regarding the variability in fatigue tasks and the lack of consistency in task movements, and include appropriate references for this point.

Reviewer #2: There are some parts which need reconsider and revision in order to strengthen the manuscript.

The main point discussion of gender effect needs to clarify and focus. Besides, subgroup analysis of one-leg or two legs landing is also interesting.

Please see the attached file

6. PLOS authors have the option to publish the peer review history of their article (what does this mean? ). If published, this will include your full peer review and any attached files.

**Do you want your identity to be public for this peer review?** For information about this choice, including consent withdrawal, please see our Privacy Policy .

Reviewer #1: **Yes: ** Makoto Asaeda

Reviewer #2: No

---

## [Author Response · Author response to Decision Letter 1]

19 Feb 2025

List of Responses

Dear Editors and Reviewers: We would like to express our gratitude for the time and effort you dedicated to reviewing our manuscript entitled “Gender Differences in the Impact of Fatigue on Lower Limb Landing Biomechanics and Their Association with Anterior Cruciate Ligament (ACL) Injuries: A Systematic Review and Meta-Analysis” (PONE-D-24-38651). Your insightful comments have significantly improved the quality of our work. On behalf of my co-authors, we thank you for allowing us to revise our manuscript. These comments are all valuable and very helpful for revising and i mproving our paper, and they provide important guiding significance to our research. We have studied the comments carefully and have made corrections, which we hope meet your approval. Revised portions are marked in red in the paper. Please find below our point-by-point responses to your feedback.

Editor:

1.Response to comment:

Response

We have made adjustments to the manuscript in accordance with the format requirements of PLOS ONE, including file naming and formatting changes. We have referred to the format template you provided.

2.Response to comment:

Please ensure that your PRISMA flow diagram is included in your main manuscript file as Figure 1 which is now in Figure 2;

Response

We have moved the PRISMA flowchart from the original "Figure 2" to "Figure 1" and modified it according to the guidelines from PLOS ONE. Thank you for your reminder!

3.Response to comment:

We note that your Data Availability Statement is currently as follows: [All relevant data are within the manuscript and its Supporting Information files.]

Response

In the manuscript, we expressly state that all relevant data has been included in the manuscript and its supplementary information file.

4.Response to comment:

Please amend either the title on the online submission form (via Edit Submission) or the title in the manuscript so that they are identical.

Response

We have ensured that the online submission form and the title in the manuscript are consistent.

5.Response to comment:

Please upload a copy of Supporting Information Figure/Table/etc. S1 to S11 which you refer to in your text on pages 44 and 45.

Response

We have uploaded all the supporting information charts and tables mentioned in the text, ensuring they are consistent with the citations, and made some minor revisions.

6.Response to comment:

Line 187: (Table3)Lower than other studies, should be sub-analysis?

Response

Thank you for your valuable suggestion. You are absolutely right that age can affect the reliability of results in a conventional meta-analysis. However, in the context of this study, conducting a separate subgroup analysis for this particular study may lead to inaccurate results due to the limited sample size. After performing a leave-one-out sensitivity analysis, we found that the robustness and reliability of the results were not affected, indicating that the sensitivity analysis supports the current findings and discussion. Based on this, including this study contributes to increasing the overall sample size, thereby enhancing the validity and reliability of the results.

That being said, we truly appreciate your suggestion. If you believe that excluding this study would be a more appropriate approach, we would be happy to further discuss possible solutions, including subgroup analyses or other methods that could strengthen the robustness of our findings.

Reviewer #1:

Thank you for your recognition and insightful comments on our research work!We will refer to your suggestions for further revision of the paper to meet the requirements of journal publication.

1.Response to comment:

Line 98-99: Please describe the method used to ensure agreement among investigators (e.g., whether recommendations by grade were applied).

Response

Thank you for your advice. We have added to the manuscript methods on how to ensure researcher consistency. Add as follows:

The systematic search was conducted by two independent researchers (C.L., W.Q.). Initially, articles were screened based on titles and abstracts. If the information was unclear or ambiguous, the full texts were retrieved for further assessment. Following this, the researchers compared their findings and resolved any discrepancies through consensus discussions. In cases of unresolved disagreement, a third independent reviewer (W.P.) was consulted to provide a final decision.

To ensure agreement among the investigators, a rigorous process was followed:

1.Predefined Eligibility Criteria: All researchers applied a set of clearly defined inclusion and exclusion criteria, which were established prior to the screening. These criteria covered study design, population characteristics, interventions, and outcomes.

2.Independent Screening: Each researcher independently screened the articles at both the title/abstract and full-text stages based on the predefined criteria.

3.Consensus Process: After the initial screening, any disagreements regarding study eligibility were addressed through consensus discussions. Studies that could not be agreed upon were jointly reassessed by all researchers to reach a final decision.

4.Third-party Review: When discrepancies persisted or were particularly complex, a third independent reviewer (W.P.) , with expertise in the subject area, was consulted to resolve the issue.

5.Quality Assessment Using STROBE Guidelines: The quality of the included studies was evaluated using a modified version of the STROBE guidelines, which assessed key methodological aspects such as study design, data collection, and potential biases.

2.Response to comment:

Line 173: Yes/No responses for items 1-10 in Table 2 are missing.

Response

We have corrected the missing data in Table 2 to ensure that all yes/No questions have full answers. Add as follows:

Table 3.Quality scores of studies included in the review

Study N1 N2 N3 N4 N5 N6 N7 N8 N9 N10 Overall

Abbey C. Thomas et al.(39) Y NA Y NA Y Y NA NA Y Y 6

Anne Benjaminse et al.(40) Y NA Y Y Y Y NA NA Y Y 7

Danielle M.Brazenet al.(41) Y Y Y Y Y Y NA NA Y Y 8

DAVID R. BELL et al.(42) Y Y Y Y Y Y NA NA Y Y 8

Dominic Gehring et al.(11) Y NA Y NA Y Y NA NA Y Y 6

Evangelos Pappas et al.(43) Y NA NA NA NA Y Y NA Y Y 5

Kristı´n Briem et al.(44) Y NA Y Y NA Y NA NA Y Y 6

Lessi G .C et al.(45) Y Y NA Y Y Y NA NA Y Y 7

Marijeanne Liederbach et al(21) Y NA Y Y NA Y Y NA Y Y 7

Michael P. Smith et al.(46) Y NA Y Y Y Y NA NA Y Y 7

Ram Haddas et al. (47) Y Y NA Y Y Y Y NA Y Y 8

Scott G. M et al.(48) Y Y NA Y Y Y NA NA Y Y 7

Thomas W. Kernozek et al.(12) Y Y Y Y Y Y NA NA Y Y 8

Zhang Qiang et al.(49) Y NA NA Y NA Y NA NA Y Y 5

N=No, NA= Not Applicable, Y=Yes. The scale questions are in Table 3

3.Response to comment:

Line 181: Would it be more appropriate to use "peripheral" instead of "isolated"? Similarly, consider whether "central" would be better than "whole body."

Response

We reviewed the articles I included and found that All fatigue induction protocols involved peripheral fatigue, so we changed it to "All fatigue induction protocols involved peripheral fatigue."

4.Response to comment:

Table 3: It may improve readability if you classify the fatigue protocol as either central or peripheral before providing detailed protocol information.

Response

We have labeled the fatigue protocol as "peripheral" and added more detailed information to the table to improve readability.

5.Response to comment:

Line 213-214: This content should be in the Discussion section, not in the Results section.

Response

We have moved this section from the results section to the discussion section and made the necessary changes.

6.Response to comment:

Line 319-320: The phrase "strategy to reduce ACL loading" may be better interpreted as a compensatory mechanism in response to fatigue, rather than simply reducing the load on the ACL.

Response

We have revised this statement in response to your suggestion to emphasize compensation mechanisms after fatigue, rather than just reducing ACL load. Changed to: “For peak knee flexion angles, both genders showed an increasing trend post-fatigue, which is considered a compensatory mechanism after fatigue and also a strategy to reduce ACL loading”

7.Response to comment:

Line 330-334: If this study does not compare athletes with non-athletes, this statement is unnecessary. Furthermore, concluding that individual responses vary could undermine the overall meaning of this review, so caution is advised.

Response

We have removed the section on how athletes compare to non-athletes.

8.Response to comment:

Line 342: As mentioned earlier, this is not about reducing the risk of ACL injuries through fatigue-induced behavior, but rather a compensatory movement in response to fatigued muscles. It's also important to note that slight knee flexion does not necessarily increase the risk of ACL injury; in fact, mild flexion could relax the MCL and LCL, potentially contributing to rotational instability. Please revisit this point from the Introduction onward.

Response

We have reviewed and revised this section to ensure that it is in line with the current understanding, and amended it to read: “The human body's inherent ability to adapt and regulate enables it to modify lower limb kinematics, thereby reducing the risk of ACL injuries(57). This can be considered a natural protective mechanism against landing impacts. However, while knee flexion can help mitigate impact forces to some extent, improper flexion may compromise the stability of the medial collateral ligament (MCL) and lateral collateral ligament (LCL), resulting in rotational instability of the knee and an increased risk of ACL injuries(25). This observation further elucidates why ACL injuries frequently occur during the landing phase of basketball(58).”

9.Response to comment:

Line 350-351: Reference 58 reflects an in vivo perspective, and while increased quadriceps contraction can increase tension on the ACL, it also suppresses rotational stress. Please take this fact into account in your discussion.

Response

We have taken into account the points in reference 58 in our discussion, in particular the effect of increased quadriceps contraction on ACL load, and explained its role in reducing rotational stress. “Increased quadriceps activation during landing preparation might increase ACL loading(63). However, this activation also plays a critical role in reducing rotational stress on the knee joint. In fact, the protective effect of quadriceps activation against rotational instability can, to some extent, offset its contribution to increased ACL tension.”

10.Response to comment:

Line 368-375: While the content is not incorrect, if this is not a review focusing on electromyography, this discussion may be unnecessary.

Response

At your suggestion, we have removed the excessive discussion of EMG to ensure that the discussion focuses on the core content of this study.

11.Response to comment:

Line 416: Please add a limitation regarding the variability in fatigue tasks and the lack of consistency in task movements, and include appropriate references for this point.

Response

We have added limitations on fatigue tasks and motion variability to the limitations section. Add as follows

The study has several limitations that should be considered. First, although 14 studies were included, only a few reported ankle biomechanical variables, which limited the scope of the analysis. This restriction hinders the investigation of the impact of ankle biomechanics on lower limb kinematics and ACL injury risk during landing. Second, the review focused solely on lower limb biomechanical indicators, without considering the positioning and control of the trunk and the entire kinetic chain, which could significantly influence knee biomechanics(74). Neglecting potential interactions across the kinetic chain, including trunk and hip control, may lead to an incomplete understanding of knee biomechanics. This limitation may underestimate the interdependencies between body segments and their effect on ACL injury mechanisms. Third, while the study emphasized gender, it did not perform a subgroup analysis of the effects of different landing styles and fatigue protocols on lower limb biomechanics. Variations in landing styles and fatigue protocols could alter the biomechanics of lower limb joints during landing. Finally, although the 14 studies included in this review reported data on both sexes before and after fatigue, studies focusing on a single sex were excluded. This omission may affect the comprehensive understanding of gender differences. Future research should address these limitations by incorporating a broader range of biomechanical variables, conducting subgroup analyses, and including sex-specific studies to provide a more comprehensive understanding of ACL injury mechanisms and prevention strategies.

Reviewer #2:

Thank you for your recognition and insightful comments on our research work!We will refer to your suggestions for further revision of the paper to meet the requirements of journal publication.

1.Response to comment:

There are some parts which need reconsider and revision in order to strengthen the manuscript.

Response

We have clarified the discussion of gender effects and focused more on their specific impact in research. The relevant sections have been reorganized to improve their science and clarity.

2.Response to comment:

The main point discussion of gender effect needs to clarify and focus. Besides, subgroup analysis of one-leg or two legs landing is also interesting.

Response

This suggestion is very useful, but since the purpose of this study is to explore gender differences, I have already written this issue in the part of limitations, and I will follow your suggestions in the later research.

---

## [Decision Letter · Decision Letter 1]

14 Mar 2025

Gender Differences in the Impact of Fatigue on Lower Limb Landing Biomechanics and Their Association with Anterior Cruciate Ligament (ACL) Injuries: A Systematic Review and Meta-Analysis

PONE-D-24-38651R1

Dear Dr. Li,

We’re pleased to inform you that your manuscript has been judged scientifically suitable for publication and will be formally accepted for publication once it meets all outstanding technical requirements.

Kind regards,

Alessandro Mengarelli

Academic Editor

PLOS ONE

Additional Editor Comments (optional):

Reviewers' comments:

Reviewer's Responses to Questions

**Comments to the Author**

1. If the authors have adequately addressed your comments raised in a previous round of review and you feel that this manuscript is now acceptable for publication, you may indicate that here to bypass the “Comments to the Author” section, enter your conflict of interest statement in the “Confidential to Editor” section, and submit your "Accept" recommendation.

Reviewer #1: All comments have been addressed

2. Is the manuscript technically sound, and do the data support the conclusions?

Reviewer #1: Yes

3. Has the statistical analysis been performed appropriately and rigorously? 

Reviewer #1: Yes

4. Have the authors made all data underlying the findings in their manuscript fully available?

Reviewer #1: Yes

5. Is the manuscript presented in an intelligible fashion and written in standard English?

Reviewer #1: Yes

6. Review Comments to the Author

Reviewer #1: Thank you for the appropriate corrections. I have requested minor corrections, so I will accept it. Thank you for reporting your valuable research results.

7. PLOS authors have the option to publish the peer review history of their article (what does this mean? ). If published, this will include your full peer review and any attached files.

**Do you want your identity to be public for this peer review?** For information about this choice, including consent withdrawal, please see our Privacy Policy .

Reviewer #1: No

---

## [Editor Report · Acceptance letter]

PONE-D-24-38651R1

PLOS ONE

Dear Dr. Li,

I'm pleased to inform you that your manuscript has been deemed suitable for publication in PLOS ONE. Congratulations! Your manuscript is now being handed over to our production team.

Kind regards,

on behalf of

Dr. Alessandro Mengarelli

Academic Editor

PLOS ONE